# Mass Spectrometry-Based Proteomic Profiling of a Silvaner White Wine

**DOI:** 10.3390/biom13040650

**Published:** 2023-04-04

**Authors:** Wendell Albuquerque, Parviz Ghezellou, Leif Seidel, Johannes Burkert, Frank Will, Ralf Schweiggert, Bernhard Spengler, Holger Zorn, Martin Gand

**Affiliations:** 1Institute of Food Chemistry and Food Biotechnology, Justus Liebig University Giessen, Heinrich-Buff-Ring 17, 35392 Giessen, Germany; 2Institute of Inorganic and Analytical Chemistry, Justus Liebig University Giessen, Heinrich-Buff-Ring 17, 35392 Giessen, Germany; 3Department of Beverage Research, Geisenheim University, Von-Lade-Strasse 1, 65366 Geisenheim, Germany; 4Institute for Viticulture and Oenology, Bavarian State Institute for Viticulture and Horticulture (LWG), An der Steige 15, 97209 Veitshöchheim, Germany; 5Fraunhofer Institute for Molecular Biology and Applied Ecology, Ohlebergsweg 12, 35392 Giessen, Germany

**Keywords:** Silvaner, proteomics, wine, proteins, mass spectrometry, *Vitis vinifera*

## Abstract

The comprehensive identification of the proteome content from a white wine (cv. Silvaner) is described here for the first time. The wine protein composition isolated from a representative wine sample (250 L) was identified via mass spectrometry (MS)-based proteomics following *in-solution* and *in-gel* digestion methods after being submitted to size exclusion chromatographic (SEC) fractionation to gain a comprehensive insight into proteins that survive the vinification processes. In total, we identified 154 characterized (with described functional information) or so far uncharacterized proteins, mainly from *Vitis vinifera* L. and *Saccharomyces cerevisiae*. With the complementarity of the two-step purification, the digestion techniques and the high-resolution (HR)-MS analyses provided a high-score identification of proteins from low to high abundance. These proteins can be valuable for future authentication of wines by tracing proteins derived from a specific cultivar or winemaking process. The proteomics approach presented herein may also be generally helpful to understand which proteins are important for the organoleptic properties and stability of wines.

## 1. Introduction

The white grape Silvaner (synonym Grüner Silvaner) is an autochthonous cultivar from Austria, originating from a genetic crossing of the cultivars Traminer and Österreichisch-Weiß [1]. Being only marginally important in today’s Austria, the grape variety is of highest importance in the region of Franconia (Franken, in German), where it was introduced at the end of the seventeenth century. Thus, Silvaner can be considered as a very old grape variety [2]. In 2021, 4535 ha of vineyards in Germany are planted with Silvaner, which corresponds to 6.5% and 4.4% of the white (70,138 ha) and total wine growing area (103,421 ha) in Germany, respectively [3]. Moreover, Silvaner is cultivated in various countries, including France (Alsace), Romania, Slovakia, Croatia, Italy (Trentino-Alto Adige), Austria, the United States and Australia. Wines of the cultivar are generally characterized to have mild acidity and subtle aromas. The on-going climate change has also been shown to affect the quality of Franconian Silvaner wines, particularly increasing sugar levels and decreasing acidity, thereby altering the wines’ sensory characteristics [4]. Furthermore, increased temperatures and decreased precipitation amounts, a frequent consequence of climate change in many wine growing regions, increased the risk for protein haze formation in the wine [5]. Proteins that survive the vinification process can interact with other wine components (e.g., ethanol) to influence the wine aroma, flavor, texture astringency, and color [6]. Additionally, wine proteins and their interactions with other wine components affect the product stability [7,8] and foaming properties [9]. Although wine proteins represent only minor components of wines, they can act as antioxidants by interacting with polyphenols [10] and some of them are likely to be allergens [11]. In addition, the remaining proteins may contribute to wine authentication by providing information about the winemaking process [12] and grape cultivation [13].

Most wine proteins originate from the plant *Vitis* spp. (less abundant fractions are derived from fermentative organisms or parasites), and therefore, factors such as soil conditions, weather and plant stress can influence the wine proteome [14,15]. Moreover, it has been discussed that the state of maturity of the grape berries highly influences the efficiency of the protein expression [16,17]. The total wine protein content also depends on a plethora of different and variable processing unit operations during harvest and in the winery [18,19]. For example, the protein concentrations of Silvaner wines from a single winery varied over four consecutive years from rather low to high levels (0.10–0.22 mg/L) compared to other wines (0.03–0.26 mg/L) [20]. In addition, proteins from microorganisms, typically from the yeast *Saccharomyces cerevisiae* [21] or grape pathogens, such as *Botrytis cinerea* [9,22], have been reported to survive the vinification process. Further proteins, such as casein, lysozyme, gelatin, and isinglass may be applied as clarification or preservation agents and may partly be transferred into the bottled wines [23]. In brief, the wine proteome is expected to be highly diverse. Among all grape proteins, a major research focus is on thermolabile proteins, such as thaumatin-like proteins (TLPs) and chitinases (CHIs), which are assumed to be responsible for major economic losses through their key role as wine haze promoters [24,25].

In the last decades, mass spectrometry-(MS)-based proteomics has evolved as a powerful research technology that has also been exploited in oenology [12]. MS techniques based on liquid chromatography coupled to electrospray ionization (LC-ESI-MS) and matrix-assisted laser desorption ionization (MALDI)/-time of flight (TOF) have been successfully applied for the characterization of proteins of different wine varieties such as Chardonnay, Semillon, Sauvignon blanc, Pinot noir and others [13]. For example, Flamini and de Rosso [26] applied MALDI-TOF for the identification of *V. vinifera* grape varieties and tissue extracts. High resolution (HR)-MS-based proteomics analysis has provided advances in terms of accurate protein identification and enough sensitivity to study even low abundance species [27]. However, this potential has not yet been fully exploited in studies on wine proteomes and applications of recent advances in MS on wine research are still emerging [28].

Proteomics commonly refers to the mass spectrometric identification and sometimes quantification of the comprehensive set of proteins present in a system [29]. Complementary sample preparation steps, such as chromatography, one dimensional (1D) or two dimensional (2D) electrophoresis, dialysis, ultrafiltration, isoelectric focusing and immunodetection are usually applied prior to mass spectrometric analysis [30].

In addition, protein digestion techniques, either *in-gel* or *in-solution*, are routinely applied in bottom-up MS analyses before sample analyses by LC-MS [31], supporting the identification of proteins. *In-solution* digestion is a gel-free and less demanding method in terms of sample preparation, whereas the *in-gel* digestion is reported to be robust, reproducible and effective, however, being known to cause protein losses due to the fractionation of the protein mixture by gels [31]. Protein separation by LC and gel electrophoresis has often been employed in MS-based proteome analyses of wines [12,22], increasing the sensitivity (by reducing protein mixtures) and thus the number of identified proteins [26].

To date, the proteomic profile of Silvaner wine has not been reported in the literature. Here, we describe for the first time the comprehensive protein identification of a Silvaner wine using the combination of two MS-based bottom-up approaches based on *in-gel* and *in-solution* digestion. The analytical approach here described might be applied to determine protein “fingerprints” for wine authentication.

## 2. Materials and Methods

### 2.1. Chemicals

High-performance liquid chromatography (HPLC)-grade water was purchased from Thermo Fisher Scientific (Bremen, Germany). Rapigest SF surfactant was obtained from Waters (Milford, MA, USA). TRIS and TRIS-hydrochloride were obtained from Carl Roth (Karlsruhe, Germany). Ammonium bicarbonate (ABC), dithiothreitol (DTT), iodoacetamide (IAA), formic acid (FA), trifluoroacetic acid (TFA) and acetonitrile (ACN, gradient grade) were obtained from Merck (Darmstadt, Germany), while MS-grade trypsin was purchased from Promega (Madison, WA, USA).

### 2.2. Silvaner Wine

Silvaner grapes were harvested from the “Würzburger Pfaffenberg” vineyard (Würzburg, Germany) on 19 September 2018 and processed to must and wine by the Bavarian State Institute for Viticulture and Horticulture (LWG, Veitshöchheim, Germany). The pH of the must and wine sample was measured using a titrator (TitroLine alpha plus with TA20 plus, TM 125 and Titrisoft 3.1 SI Analytics, Mainz, Germany). The must had a measured weight of 99°Oe (DMATM 35, Anton Paar, Graz, Austria), a total acidity of 5.0 g/L (as tartaric acid) and a pH value of 3.5 (after adding 1.5 g/L tartaric acid to lower the pH). The grapes were not destemmed and only lightly crushed (crush roller, Scharfenberger Maschinenbau, Bad Dürkheim, Germany). The maceration time was 4 h at 16 °C. The solid-liquid separation was performed using a pneumatic, partially slotted tank press with a volume of 900 L (Europress P9, Scharfenberger Maschinenbau). Pectinase treatment was carried out at the must stage with 2 mL/hL (Trenolin Rapid, Erbslöh, Geisenheim, Germany). After enzymation, the must sedimented for 12 h at 16 °C and then the clear supernatant was drawn off and used for fermentation. For better nutrition of the yeast, 200 mL/hL of Vitamon Liquid (Erbslöh) was added as a yeast nutrient (combination nutrient of vitamin B_1_ and diammonium phosphate). The commercial yeast strain “Oenoferm Terra” (Erbslöh) was used at 20 g/hL to ferment the must for 21 days at 17 °C, while in the last third of fermentation the temperature was increased to 18 °C to obtain a safe final fermentation. The obtained wine had an alcohol content of 11.31%, fermentable sugars of 3.4 g/L, total acidity of 5.1 g/L (calculated as tartaric acid), a pH of 3.35, volatile acid content of 0.24 g/L, free SO_2_ (incling reductones) content of 102 mg/L, reductone levels of 66 mg/L, and an effective content of free SO_2_ at 36 mg/L. The bentonite (NaCalit PORE-TEC, Erbslöh) requirement, determined by a heat test (4 h at 80 °C in a drying oven (UNB 200, Memmert, Büchenbach, Germany), subsequent cooling and then evaluation with turbidity meter (Turb 430 IR, WTW, Weilheim, Germany)) was extremely high (450 g/hL), which indicated a high content in proteins and proteinaceous colloids.

### 2.3. Technical Scale Isolation and Analysis of Silvaner Wine Colloids

The ultrafiltration of the protein-rich colloid of the Silvaner wine (250 L) was performed as described by Albuquerque et al. [32]. Briefly, the wine was firstly sheet-filtered by using a stainless steel sheet filter (40 cm × 40 cm, Pall-Seitz-Schenk, Bad Kreuznach, Germany) packed with 5 filter sheets (K 250, Pall-Seitz-Schenk). Ultrafiltration was subsequently performed with a Sartocon beta system (Sartorius, Göttingen, Germany) equipped with two 0.6 m^2^ Sartocon Hydrosart cassettes with a molecular mass cut-off (MWCO) of 10 kDa. A subsequent diafiltration step, performed with citrate buffer (5 g citric acid per L, pH 4) and water, aimed to remove low molecular weight substances. However, still low molecular weight wine components bound to the colloids may remain in the isolated colloids. After the lyophilization of the retentate, the resulting powder was hygroscopic and, thus, stored in airtight containers at room temperature.

The carbohydrate content of the isolated colloids was determined by quantitation of neutral sugars and uronic acids released after hydrolysis with sulphuric acid by high performance anion exchange chromatography with pulsed amperometric detection (HPAEC-PAD) as described beforehand [32]. Additionally, the total protein content of the isolated colloids was determined after colloid hydrolysis by measuring the released amino acids by anion exchange chromatography according to Ahlborn et al. [33]. The wine colloids contained 47.1% of carbohydrates and 34.7% of protein in the dry matter. Residual moisture, determined by a moisture analyzer (ML-50, AND, Tokyo, Japan) at 120 °C with 0.5 g sample, was 8.9%. Based on the yield of the ultrafiltration and the residual moisture, the studied Silvaner wine contained 0.63 g colloid per L wine [20].

### 2.4. Protein Content and Visualization

Protein in the isolated colloid and from chromatographic runs (see Section 2.5.1) were quantified according to Bradford [34], with bovine serum albumin (Carl Roth) as standard. Proteins were separated by sodium dodecyl sulfate polyacrylamide gel electrophoresis SDS-PAGE (12% polyacrylamide gel) according to Laemmli [35] under denaturing conditions. After separation, protein spots were visualized by Coomassie blue staining (Thermo Fisher Scientific).

### 2.5. MS-Based Proteomics Analysis of Proteins from a Silvaner Wine

The aforementioned isolated colloid was submitted to size exclusion chromatography (SEC) and subsequent *in-solution* and *in-gel* digestion, as described in Figure 1.

#### 2.5.1. *In-solution* Digestion: Protein Fractionation by SEC Chromatography

The proteins present in the isolated wine colloid with 0.5 ± 0.1 mg/mL were fractionated using a HiLoad 16/60 Superdex 200 prep grade size-exclusion chromatography column (GE Healthcare Biosciences, Uppsala, Sweden) on a fast protein liquid chromatography (FPLC) system (Bio-rad NGC™ Quest Plus, Feldkirchen, Germany), using 50 mM Tris-HCl (pH 7, containing 150 mM NaCl) as eluent at 1 mL/min. Proteins were detected at 280 nm and automatically collected by a fraction collector (BioFrac™, Bio-Rad). The % of the yield from the protein fractions after FPLC fractionation is shown in Appendix A. The retention time was correlated to the molecular mass based on gel filtration protein standards (from 1350 kDa to 670,000 kDa, Bio-Rad) using the software ChromLab version 6.1.29 (Bio-Rad).

To perform the *in-solution* digestion, aliquots of 25 μL of wine proteins collected from the SEC (standardized at 1 μg/μL by vacuum concentration or dilution) were mixed with 5 µL of a 50 mM ammonium bicarbonate solution and 20 µL of a RapiGest solution (0.1% dissolved in ABC) and vortexed. Subsequently, the mixture was incubated with 5 μL of 5 mM dithiothreitol dissolved in ABC at 60 °C for 15 min. Protein alkylation was performed by incubation with 5 μL of 200 mM iodoacetamide dissolved in ABC for 30 min at 25 °C. Trypsin digestion was performed by the addition of 1.25 µL trypsin/Lys-C mix (0.5 µg/µL in ABC buffer), further incubation at 37 °C for 16 h, and then stopped by the addition of 2 μL of 100% formic acid. The samples were centrifuged (15 min at 4 °C and about 13,000× *g*) and concentrated using a vacuum concentrator (Eppendorf, Hamburg, Germany). The obtained digestates were resuspended in 100 μL of ultrapure water, desalted by ZipTip C18 pipette tips (Merck), vacuum concentrated and stored for further analysis.

#### 2.5.2. *In-gel* Digestion: Proteins Fractionated by Gel Electrophoresis

Proteins were further separated by SDS-PAGE based on their molecular mass, as described in Section 2.4. After protein separation, the bands were excised from the gels with a scalpel and the gel pieces were subsequently supplemented with 30 μL of 50% ACN for 15 min, 20 μL of 0.1 M ABC solution for 5 min and 30 μL of a 100% ACN solution for 15 min. After vacuum concentration, the gel pieces were incubated in 50 μL of a 10 mM DTT solution (dissolved in 0.1 M ABC solution) for 45 min at 56 °C, 30 μL of a solution of 55 mM iodoacetamide (in 0.1 M ABC) for 30 min at 25 °C and 20 μL of a 0.1% RapiGest solution (dissolved in 50 mM ABC solution) for 30 min at 37 °C. The gel pieces were dried again and a trypsin solution (0.5 µg/µL solved in 50 mM ABC) was added for protein digestion for 16 h at 37 °C. Afterwards, the samples were centrifuged (13,000× *g*, 10 min, 4 °C) and the supernatants were used for further analysis.

#### 2.5.3. Liquid Chromatography Mass Spectrometry (LC-MS) Analysis

The digested peptides were separated using a UHPLC system (UltiMate 3000 RSLC HPLC system, Ultra-High-Performance Liquid Chromatography, Thermo Fisher Scientific). A Kinetex C18 (2.1 mm × 100 mm, 2.6 µm 100 Å particle size) column (Phenomenex, CA, Torrance, USA) was used to separate the digests at a flow rate of 250 µL/min following an optimized gradient of the solvents A (aqueous 0.1% (*v*/*v*) water) and B (ACN/0.1% formic acid): isocratic flow (2% B) for 5 min, followed by a gradient of 2–40% (B) for 70 min, 40–50% (B) over 5 min and 50–98% (B) for 2 min. Re-equilibration was obtained by an isocratic flow at 2% of B for 10 min. The HPLC system was coupled to a Q Exactive HF-X (Thermo Fisher Scientific) mass spectrometer. The MS device was operated in data-dependent acquisition (top-10 DDA) mode with the following parameters for full MS scans: mass range of *m/z* 350 to 1800, resolution of 120,000 (at *m/z* 200), automatic gain control (AGC) target of 3 × 10^6^, injection time (IT) of 50 ms; and MS/MS scans: mass range of *m/z* 200 to 2000, mass resolution of 30,000 (at *m/z* 200), AGC target of 1 × 10^5^, IT of 120 ms, isolation window *m/z* 1.3 and dynamic exclusion duration set to 60 s.

#### 2.5.4. MS Data Analysis

Protein sequences were obtained through shotgun searching performed by the software Proteome Discoverer (PD) version 2.4 (Thermo Fisher Scientific). The organisms *Vitis vinifera* and *Saccharomyces cerevisiae* were taxonomically set for the search. Protein sequences from both organisms were downloaded from the UniProt protein database [36] and used as a personal database. Other organisms, which are pathogens or participate in the fermentative process, were included in the database search (see Section 3.2., i.e., the methylotrophic bacterium *Methylobacterium* sp., which has epiphytic interactions with grapes and can survive during the wine production [37]). The peptide precursor and fragment ion mass tolerance in PD were set to 10 and 0.5 ppm, respectively. The parameters were assigned to a maximum of two missed cleavage sites of trypsin digestion and a minimum peptide length of 6. The dynamic modification was set to an oxidation (+15.995 Da (M)) and static modification to carbamidomethyl (+57.021 Da (C)). Percolator node was used to validate the identified peptide-spectrum matches (PSMs) and filter the data with parameters of a strict target FDR (false discovery rate) of 0.01 and a relaxed target FDR of 0.05. The MaxQuant contaminant database was used to mark the contaminants in the results file and proteins with at least one identified unique peptide were considered in the survey. “Characterized” proteins were considered those with annotated functional information in the database.

## 3. Results

### 3.1. Protein Fractionation and Visualization

Proteins (Figure 2a) separated by size exclusion chromatography (SEC) were collected in four main fractions (A, B, C and D), with the proteins represented by the largest peak in the range of 20–70 kDa and collected in fraction C. The collected proteins from each chromatographic peak were subjected to *in-solution* digestion bottom-up MS-based proteomics and were further separated according to their molecular mass (also described as molecular weight (MW)) by SDS-PAGE, resulting in a total of 16 protein bands (Figure 2b). Fraction A from SEC showed a single protein band greater than 170 kDa, fraction B showed two bands between 130 and 55 kDa, fraction C showed the densest protein bands, with a total of 12 spots from 72 to 20 kDa and finally fraction D revealed two bands from 17 to 10 kDa.

### 3.2. MS-Based Proteomics Analysis

A total of 154 proteins (with different identification numbers, but not 154 proteins with different functions) were identified by combining the data obtained from the *in-solution* and *in-gel* protein digestion methods. The identified proteins were further classified as “characterized” (with characteristics or functions described in the database) and “uncharacterized” (when no properties or functions were found in the database). Among these proteins, 88 were only identified with the *in-gel* digestion method (48 characterized and 40 uncharacterized), while 45 other proteins were exclusively found with the *in-solution* digestion approach (38 characterized and seven uncharacterized). Moreover, 21 further proteins were commonly found after both digestion methods (16 characterized and five uncharacterized) (Figure 3). Table 1 (characterized) and Table 2 (uncharacterized) list all identified proteins, according to the respective digestion method applied. Some proteins were repeatedly found; therefore, only the those with the highest coverage and identified unique peptides are presented. The complete protein list is available as Appendix A. The proteins had molecular masses ranging from 6.4 to 372.2 kDa. Figure 2 shows the correlation of each spot in the gel (spots 1 to 16) with some of the identified proteins by MS proteomics analysis (*in-gel* analysis). The complete list of identified proteins for each gel spot (Figure 2) is available in the Appendix A. The organism source and MW for each protein are given and the characterized proteins have a description associated with their accession numbers. Proteins from 10 additional organisms were included in the database of *Saccharomyces cerevisiae*, because they are eventually found as grape pathogens or fermentative organisms. Among them, we identified proteins from *Ashbya gossypii* (*n* = 5), *Cyberlindnera fabianii* (*n* = 4), *Kazachstania saulgeensis* (*n* = 2), *Methylobacterium* sp. (*n* = 2), *Novosphingobium* sp. (*n* = 2), *Pichia kudriavzevii* (*n* = 2), *Geotrichum candidum* (*n* = 1), *Aspergillus niger* (*n* = 1) and *Penicillium citrinum* (*n* = 1).

## 4. Discussion

With the availability of high-throughput and rapid screening methods and HR-MS techniques, the evaluation of wine processing and an overview of the metabolism and defense mechanisms of grapes are feasible [26]. Therefore, MS-based proteomics may be applied to authenticate wines as a “proteome signature” to avoid fraudulent products in the wine market [12] in addition to other methods such as polyphenolic profiling (based on HPLC coupled with ultraviolet (UV) and MS analysis (HPLC-UV-MS/MS)) [40] and fluorescence spectroscopy [41]. The proteomics data reported here might serve in the future (after authenticity requirements) for a comparative authentication of Silvaner wine based on identifying particular proteins. A comparative analysis of wine proteomes showed that some proteins are commonly reported, and generally present across different cultivars. These include proteins from the vine plant *V. vinifera* (TLPs, CHIs, vacuolar invertase, (1,3)-*β*-glucanase, lipid transfer protein), from fermentative organisms, i.e., *S. cerevisiae* (acid phosphatase, seripauperin, protein YGP1, glycosidases, protein Tos1p, daughter-specific expression-related protein, and cell wall proteins) and from grape pathogens such as *A. niger* (pectin lyase).

Eventually, the reported proteins might be useful for a comparative analysis between cultivars (similarly to the analyses presented in the Table 1 and Table 3) and therefore, protein matches with at least one unique peptide were considered in the present study. In this study, the combination of two different protein fractionation steps, the HR-MS analysis and the complementary *in-solution* and *in-gel* digestion techniques allowed for a high-score level of identified proteins. In total, from the 154 proteins identified from a Silvaner wine, 80% originated from *V. vinifera* and *S. cerevisiae*, and roughly 20% from other organisms, which are frequently found to be associated with wine and grapes (Figure 4a). Protein species, which can survive the vinification process may influence the wine organoleptic properties and haze formation in wines [17]. Similar compositions of proteins from different organisms have been reported in the literature. However, the methods and the HR-MS analysis in this study provided a higher number of identified proteins compared to other studies (Table 3). Marangon et al. [38] combined hydrophobic interaction chromatography with reversed-phase liquid chromatography using HPLC and nano-LC-MS/MS analyses to improve the protein purity and the quality of the proteomics analysis of Semillon grape juice and wine. The *in-gel* digestion allowed the identification of proteins after an additional step of separation (gel electrophoresis) and had the advantage of reducing the mixture of proteins that are digested by trypsin and further fragmented during the MS analysis. However, some proteins were still detected in unexpected molecular masses (Appendix A). The number of identified proteins after *in-gel* digestion was higher than that after the *in-solution* method, which was also observed by Choksawangkarn et al. [31]. In contrast, the *in-solution* approach allowed the direct LC-MS/MS analysis of the digested peptide mixtures, avoiding the risk of protein losses during further fractionation steps. Approximately one-third of the identified proteins in this present study were exclusively found using the *in-solution* digestion method. Additionally, methods of protein extraction are compared in Table 3. Sample isolation such as the MWCO of membranes, precipitation method and pellet resuspension can reduce the final protein content and influence the proteome analysis.

The identification of low-abundance proteins originated from eventual grape infections, contaminations, distinct fermentative organisms and others are difficult to reproduced in different wine analyses, even if these grapes are from the same cultivar. The eventual presence of organisms such as pathogens [37,42], fermentative bacteria or yeasts [17,43] and factors such as differential gene expression induced by abiotic and biotic stress including climatic aspects [44,45] or protein contaminants [46,47] can greatly influence the variability of the proteomic analysis of wine. Righetti et al. [48] discussed that the wine composition and age might be affected by the presence of additives and, therefore, low-abundance proteins can evidence the vinification process. In addition, proteins from the fermentation process or added as fining agents such as egg white, as potential allergens, can influence the protein composition and may participate in the formation of haze particles [49]. The proteomics of wines has already been established as a tool for product authentication and avoiding food fraud. Ortea et al. [50] highlighted that not only vintages or cultivars, but also protein additives could be traced and characterized by proteomics analysis. Since such proteins were not identified, their absence in the clarification process of the analyzed Silvaner wine was confirmed.

Table 4 shows the classification of the characterized proteins based on their cellular functions. In total, eleven proteins were related to gene regulations and nucleotide metabolism: eight, five, and four proteins were described as participating in the metabolism of carbohydrates, proteins and lipids, respectively. Six proteins were identified as participating in the cell defense of *V. vinifera* and *S. cerevisiae*, including the pathogenesis-related TLPs and CHIs. Six proteins were related to cell structural functions, and 14 proteins (the most abundant group) are responsible for metabolic and cell signaling functions. Several proteomics studies have classified wine proteins in different classes, including the proteins involved in sugar metabolism (such as vacuolar invertases) and in stress response or plant defense (such as the pathogenesis-related proteins TLPs, CHIs and osmotin-like proteins) as well as proteins from yeast and other fungal origins [22]. In general, the distribution of the proteins of berries is known to vary with the stages of their development. In late growth stages (i.e., at full maturity, during harvesting periods), an increase in the levels of proteins involved in stress response, metabolism, plant defense, and cytoskeleton formation is significant [51].

A graphical comparison of the number of identified proteins (classified by their cellular functions) and the digestion method used (*in-gel*, *in-solution*, and *in-gel*+*in-solution*) is presented in Figure 4b. In our findings, the highest number of proteins was associated with basic cellular functions related to metabolism and cell signaling. According to Kuang et al. [51], such protein profiles are more related to late stages of berry development, which is in agreement with the fact that wines are produced from ripe fruit. Proteins related to basic cellular functions were also found by Marsoni et al. [52], when they isolated and identified 15 proteins from different grape tissues and verified that most of them were involved in the regulatory and secondary metabolism such as energy metabolism. The classes of proteins or enzymes participating in the metabolism of proteins, nucleotides and lipids were also well represented in our findings. Sarry et al. [53] identified 67 proteins from six *V. vinifera* grape varieties and classified the proteins by their cellular functions: 34% of them were involved in energy metabolism, 19% had functions in the cell defense and in the response to stress, while 13% participated in the primary metabolism.

Particularly important for the deleterious haze formation are the pathogenesis-related (PR) proteins, which exert defensive functions in diverse plant species [17]. In *V. vinifera*, they are commonly expressed on a basal level during ripening or mechanical stress, while their expression level is upregulated during plant infection [54]. The highest fraction of these PR proteins is represented by TLPs and CHIs [18,45]. These two protein species are often reported as the main contributors for haze formation and wine instability [8,55]. Many isoforms of heat unstable proteins (HUPs), such as TLPs and CHIs, as well as other proteins such as *β*-glucanases [56] are also involved in haze formation and they are often reported to have molecular masses in the range of 20–30 kDa [17].

We previously used top-down proteomics to detect peptides obtained by tryptic digestion of the same proteinaceous substance studied herein [32]. A total of nine proteins (including high and low-abundances) from our earlier study could be identified in the present study (Appendix A). Kwon [30] found a total of 20 proteins from a Sauvignon blanc wine by nano-LC-MS analysis. From these, five proteins were from grape, twelve from yeast, two from bacteria and one of fungal origin. The author emphasized that the MS analysis provided a sensitive and selective analysis for the protein identification. Okuda et al. [57], for example, detected vacuolar invertases (with a MW of approximately 66 kDa) and a lipid transfer protein (LTP, with 13 kDa) in Chardonnay wines by sequencing the *N*-terminal amino acid sequences of protein spots from 2D electrophoresis gels (electroblotted onto a Polyvinylidene fluoride (PVDF) membrane). Although the authors found approximately 150 protein spots on a 2D electrophoresis gel, most of which were related to TLP, osmotin-like protein, invertase, LTP, and their hydrolysis products. As expected, yeast proteins were also often reported as part of the wine proteome. Cilindre et al. [22] reported ten different proteins in a wine from healthy grapes and eight different proteins in a wine from grapes infected with Botrytis sp. (two protein bands probably secreted by *B. cinerea*), including a cell-wall mannoprotein from *S. cerevisiae* and two pectinolytic enzymes from *Botryotinia fuckeliana* (teleomorph of *B. cinerea*).

Proteomic profile might be comparatively used to detect differences in products from different wineries and years and validate authentication marker proteins. Proteins such as TLP, CHI, vacuolar Invertase, and protein Ygp1, detected in the Silvaner wine, are regularly found in other wine samples. Other low-abundance proteins identified in this study could be characterized as protein markers from now on. Some examples could be a cysteine proteinase inhibitor (A5ANX3) and a plasma membrane ATPase (A0A438EWP8), which are originated from the plant *V. vinifera* (to evidence a protein from the cultivar Silvaner and not from fermentative organisms), they were found here with three and two unique peptides (respectively) and were not previously identified in literature-reported wine proteomics. However, to validate the hypothesis that these proteins may be used as qualitative markers, several wines from different cultivars and geographical regions and years have to be analyzed by the same method described herein. A comparison of proteins reported from different white wines, which were also identified in the present study, can be found in the Appendix A. Rešetar et al. [58] emphasized the increase in fraud on the wine market in recent years and discussed the need for guidelines and laws to regulate standard production procedures and ensure quality parameters such as geographical origin. Chambery et al. [59] presented the concepts of food traceability based on the EU General Food Law Regulation as a form to guarantee food quality and safety. Recent advances and the availability of MS techniques could be applied in the proteomics analyses of different wines and become a powerful tool to provide information about food additives, allergenic proteins, fining agents, and haze potential to validate products and prevent commercial counterfeiting. Such methods are also recommended for the validation of suitable marker proteins based on the evaluation of many different vineyards, cultivars, years, drought, grape pathogens, and plant stress conditions.

## 5. Conclusions

The two-step protein fractionation and subsequent HR-MS techniques allowed the analysis of the comprehensive proteome profiling of a Silvaner wine for the first time. In addition, combining *in-solution* and *in-gel* protein digestion techniques enabled sufficient sensitivity to detect a high number (154 different accession numbers) of identifiable proteins. The functions of 50 proteins were described and classified according to their roles in cell metabolism, signaling, defense and structure. Such a combination of methods can improve the characterization of wine proteomes and be helpful to obtain traces of wine’s origin and processing as an authentication method for future applications.

## Figures and Tables

**Figure 1 biomolecules-13-00650-f001:**
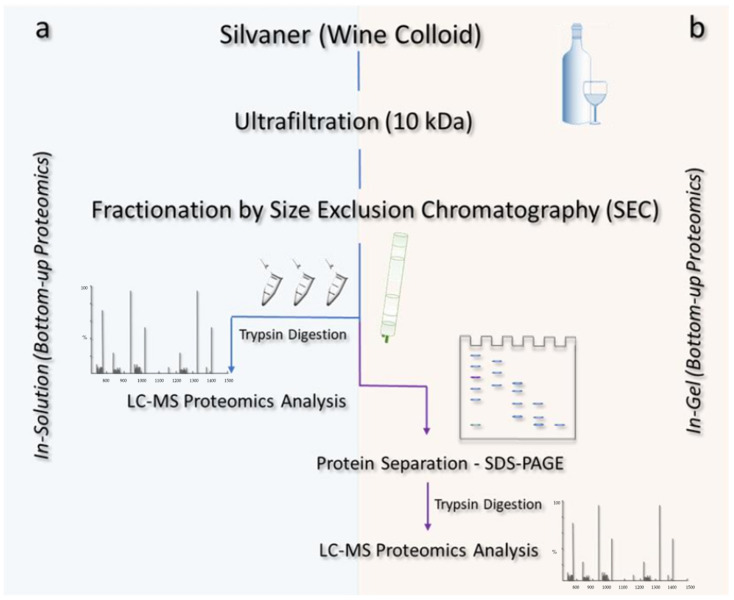
Illustrative scheme of the methods applied for the isolation and identification of proteins from a Silvaner wine. After fractionation via size exclusion chromatography (SEC), the wine proteins were subjected to distinct methods of digestion: (**a**) *in-solution*, in which the samples were directly tryptically digested and submitted to LC-MS analyses after the SEC fractionation step; and (**b**) *in-gel*, whereby the proteins were further fractionated by SDS-PAGE and then tryptically digested prior to the LC-MS analysis.

**Figure 2 biomolecules-13-00650-f002:**
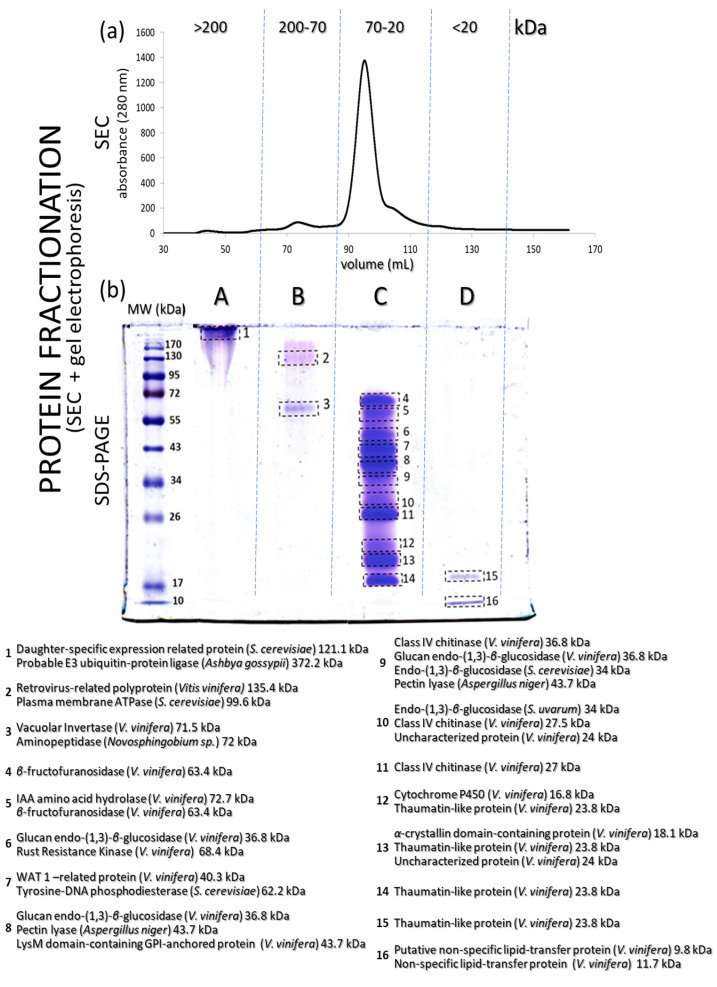
(**a**) SEC chromatogram of proteins from a Silvaner wine (separated according to molecular mass). (**b**) SDS-PAGE profile of the four main protein fractions obtained from the SEC chromatographic step shown in (**a**). Some of the identified proteins (sorted by molecular mass) are described in (**b**).

**Figure 3 biomolecules-13-00650-f003:**
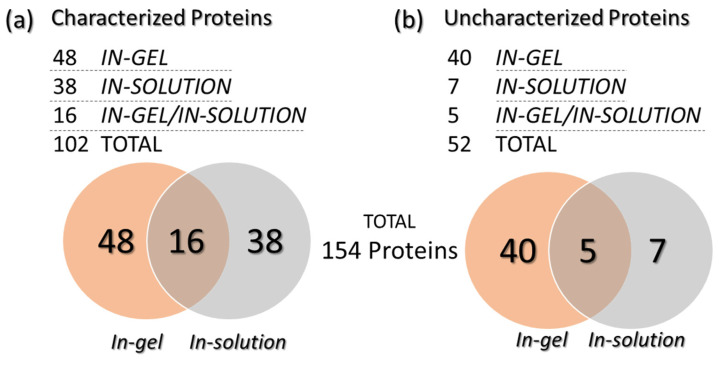
Venn diagrams presenting the number of characterized (**a**) and uncharacterized (**b**) proteins identified after *in-gel* or *in-solution* digestion.

**Figure 4 biomolecules-13-00650-f004:**
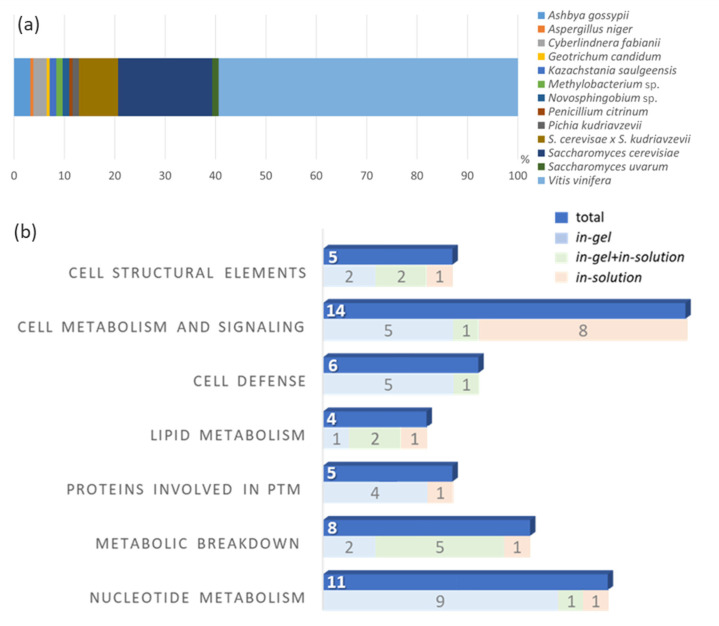
(**a**) Stack-bar blot of the percentage distribution of the found protein to the organisms (**b**) Quantitative comparison of the identified proteins from a Silvaner wine divided per cellular function. The number of proteins identified by each digestion technique is also presented. PTM means post-translational modifications.

**Table 1 biomolecules-13-00650-t001:** Characterized proteins identified by MS-based proteomics of a colloid isolated from a Silvaner wine.

	IN-GEL (Exclusively Identified by *in-gel* Digestion)					
	Accession	Gel Band	Description	Organism	MW (kDa)	Reported by (Ref *)
1	C8ZG69	1	Ygp1p	*Saccharomyces cerevisiae*	37.3	5
2	G2WD47	1	K7_Spt2p	*Saccharomyces cerevisiae*	38.5	-
3	H0GMG3	1	Ygp1p	*S. cerevisiae x S. kudriavzevii*	37.3	5
4	A0A438HVN1	1 and 12	Endochitinase EP3	*Vitis vinifera*	27.2	1,2,3,4,6
5	A0A438ENJ7	2 and 6	Retrovirus-related Pol polyprotein from transposon TNT 1-94	*Vitis vinifera*	33.7	-
6	C8Z7L9	3	EC1118_1F14_0100p	*Saccharomyces cerevisiae*	53.7	-
7	G2WEU0	3	K7_Zpr1p	*Saccharomyces cerevisiae*	55.1	-
8	A0A061ASV5	3	CYFA0S02e01574g1_1	*Cyberlindnera fabianii*	34.6	-
9	A0A1V2L9U0	3	Cytokinesis protein sepH	*Cyberlindnera fabianii*	116.3	-
10	I9C1P4	3	Aminopeptidase	*Novosphingobium* sp.	72	-
11	A0A1V2LS96	3	Putative lipase ATG15	*Pichia kudriavzevii*	56.8	-
12	A6ZPP5	5	Pathogen-related protein	*Saccharomyces cerevisiae*	30.6	-
13	C8ZFH3	5	EC1118_1M3_5204p	*Saccharomyces cerevisiae*	12.8	-
14	A0A438EI04	5 and 13	IAA-amino acid hydrolase ILR1-like 4	*Vitis vinifera*	72.7	-
15	A0A438F5Y0	5	Retrovirus-related Pol polyprotein from transposon TNT 1-94	*Vitis vinifera*	10.1	-
16	A0A438HFW8	5	UDP-glycosyltransferase 85A8	*Vitis vinifera*	20.5	-
17	A0A438HSQ5	6	Rust resistance kinase Lr10	*Vitis vinifera*	68.4	-
18	I9WYJ6	6	6-carboxy-5,6,7,8-tetrahydropterin synthase	*Methylobacterium* sp.	13.5	-
19	A0A438JNK9	7	WAT1-related protein	*Vitis vinifera*	40.3	-
20	A6ZLG3	7	Tyrosine-DNA phosphodiesterase	*Saccharomyces cerevisiae*	62.2	-
21	A6ZMC5	7	Conserved protein	*Saccharomyces cerevisiae*	104.7	-
22	A0A438C3D6	8	LysM domain-containing GPI-anchored protein 1	*Vitis vinifera*	43.7	-
23	A0A0M3M4Y7	8 and 9	Pectin lyase A	*Aspergillus niger*	39.7	5
24	O24531	8 and 11	Class IV endochitinase (fragment)	*Vitis vinifera*	27	1,2,3,4,6
25	A6ZQF9	9	Killer toxin resistant protein	*Saccharomyces cerevisiae*	30	-
26	A0A1X7QY33	9	Similar to *Saccharomyces cerevisiae* YHR098C SFB3 component of the Sec23p-Sfb3p heterodimer of the COPII vesicle coat	*Kazachstania saulgeensis*	106.6	-
27	A0A1X7R1P0	9	Similar to *Saccharomyces cerevisiae* YJL170C ASG7 protein that regulates signaling from a G protein *β*-subunit Ste4p	*Kazachstania saulgeensis*	25.7	-
28	A0A438F8T9	10	Ethylene-overproduction protein 1	*Vitis vinifera*	113.4	-
29	A0A1V2LQA7	10 and 11	Nuclear GTP-binding protein NUG1	*Pichia kudriavzevii*	58.7	-
30	A0A438F497	11	Protein HUA2-like 3	*Vitis vinifera*	187.4	-
31	H0GDF3	11	Non-specific serine/threonine protein kinase	*S. cerevisiae x S. kudriavzevii*	120	-
32	A6ZWD3	12	ATP-dependent RNA helicase DBP1	*Saccharomyces cerevisiae*	67.9	-
33	A0A438FBU5	12	Cytochrome P450 81E8	*Vitis vinifera*	16.9	-
34	A3QRB5	12, 13 and 14	Thaumatin-like protein	*Vitis vinifera*	23.9	1,2,3,4,5
35	Q75E94	13	AAR186Wp	*Ashbya gossypii*	25.8	-
36	H0GH06	13	Yor1p	*S. cerevisiae x S. kudriavzevii*	166.7	-
37	H0GRW5	13	Mak32p	*S. cerevisiae x S. kudriavzevii*	36.3	-
38	A0A438CAI5	13	Retrovirus-related Pol polyprotein from transposon RE1	*Vitis vinifera*	73.7	-
39	A0A438F753	13	5′-nucleotidase SurE	*Vitis vinifera*	39.7	-
40	A0A438KCF4	13	*α*-Crystallin domain-containing protein 22.3	*Vitis vinifera*	18.1	-
41	A0A438KHH5	13	RNA exonuclease 4	*Vitis vinifera*	44.2	-
42	A0A0J9X743	13	Similar to *Saccharomyces cerevisiae* YGL131C SNT2 DNA binding protein with similarity to the *S. pombe* Snt2 protein	*Geotrichum candidum*	153.2	-
43	I9C4L4	13	Protein ImuA	*Novosphingobium* sp.	29.1	-
44	A0A438FPT4	13	Retrovirus-related Pol polyprotein from transposon 17.6	*Vitis vinifera*	98.5	-
45	A0A1V2L627	15	Sensitive to high expression protein 9, mitochondrial	*Cyberlindnera fabianii*	42.6	-
46	H0GZX2	15	Prm1p	*S. cerevisiae x S. kudriavzevii*	73.2	-
47	A0A438IBY2	15	Retrovirus-related Pol polyprotein from transposon opus	*Vitis vinifera*	144.6	-
48	A0A438IP20	15	Putative ribonuclease H protein	*Vitis vinifera*	16.6	-
	**IN-SOLUTION (exclusively identified by the *in-solution* digestion method)**					
	**Accession**	**SEC Fraction**	**Description**	**Organism**	**MW (kDa)**	**Reported by Ref ***
49	A6ZL40	A	Acid phosphatase	*Saccharomyces cerevisiae*	52.9	1
50	B3LP15	A	Protein YGP1	*Saccharomyces cerevisiae*	37.3	5
51	A6ZM69	A	Lysophospholipase	*Saccharomyces cerevisiae*	71.6	-
52	F8KAD2	A	Exo-(1,3)-*β*-glucanase of the cell wall	*Saccharomyces uvarum*	51.2	1
53	A6ZQA6	A	Cell wall mannoprotein	*Saccharomyces cerevisiae*	29.6	-
54	A0A438EWP8	A	Plasma membrane ATPase	*Vitis vinifera*	105.8	-
55	H0GZ48	A	Lysophospholipase	*S. cerevisiae x S. kudriavzevii*	75.4	-
56	A0A438F6R5	A	Pentatricopeptide repeat-containing protein	*Vitis vinifera*	104.7	-
57	A0A438JSE9	A	Ubiquitin-60S ribosomal protein L40	*Vitis vinifera*	80.1	-
58	C7GRZ8	A	YJL171C-like protein	*Saccharomyces cerevisiae*	42.9	-
59	C8Z9T5	A	Sps100p	*Saccharomyces cerevisiae*	34.2	-
60	H0GRF2	A	Tos1p	*S. cerevisiae x S. kudriavzevii*	48.2	4
61	G2WLU7	A	K7_Ygp1p	*Saccharomyces cerevisiae*	37.3	5
62	H0GVA1	A	Glycosidase	*S. cerevisiae x S. kudriavzevii*	54.8	4,5
63	A0A438CXL6	A	Transposon Ty3-I Gag-Pol polyprotein	*Vitis vinifera*	59.1	-
64	Q753A2	A	AFR422Wp	*Ashbya gossypii*	39.2	-
65	Q758V6	A	AEL320Wp	*Ashbya gossypii*	112.9	-
66	A5ANX3	A and B	Cysteine proteinase inhibitor	*Vitis vinifera*	11.2	-
67	A0A438HVZ7	A and C	Endochitinase EP3	*Vitis vinifera*	28.6	1,2,3,4,6
68	A6ZVW2	A, C and D	Seripauperin	*Saccharomyces cerevisiae*	17.7	5
69	A0A438DZR8	A, C and D	Non-specific lipid-transfer protein	*Vitis vinifera*	11.7	3,4,6
70	A7A1R6	B	Cell wall mannoprotein	*Saccharomyces cerevisiae*	23.3	-
71	G2WE85	B	Plasma membrane ATPase	*Saccharomyces cerevisiae*	99.6	-
72	Q9P963	B	ACC synthase	*Penicillium citrinum*	48.2	-
73	A0A438J3Y1	B	Retrovirus-related Pol polyprotein from transposon TNT 1-94	*Vitis vinifera*	135.4	-
74	H0GGT5	B	Glycosidase	*S. cerevisiae x S. kudriavzevii*	53.7	1,4,5
75	C8ZED9	B	Sma2p	*Saccharomyces cerevisiae*	40.8	-
76	A6ZLA4	B and C	Target of Sbf	*Saccharomyces cerevisiae*	47.9	1
77	H0GYP4	C	Ccw14p	*S. cerevisiae x S. kudriavzevii*	25	-
78	A6ZPT3	C	GTPase-activating protein	*Saccharomyces cerevisiae*	53.9	-
79	A6ZVC9	C	Histidine kinase osmosensor that regulates an osmosensing MAP kinase cascade	*Saccharomyces cerevisiae*	134.5	-
80	H0GWM4	C	Cis3p	*S. cerevisiae x S. kudriavzevii*	23.3	-
81	H0GL37	C	Asi1p	*S. cerevisiae x S. kudriavzevii*	71.4	-
82	A0A438DEP9	C	Retrovirus-related Pol polyprotein from transposon TNT 1-94	*Vitis vinifera*	169.1	-
83	G2WJP1	C	K7_Sen1p	*Saccharomyces cerevisiae*	252.5	-
84	Q2QCI7	D	Non-specific lipid-transfer protein	*Vitis vinifera*	11.8	3,4,6
85	I9WWM7	D	PAS domain-containing protein	*Methylobacterium* sp.	21.3	-
86	Q752D0	D	AFR645Wp	*Ashbya gossypii*	44.7	-
	**IN-GEL/IN-SOLUTION (identified by *in-gel* and *in-solution* digestion)**					
	**Accession**	*Gel Band*/**SEC fraction**	**Description**	**Organism**	**MW (kDa)**	**Reported by Ref ***
87	A6ZSE1	*1/* **A**	Daughter-specific expression-related protein	*Saccharomyces cerevisiae*	121.1	1
88	C7GQJ1	*1* and *2*/**A, B**	Cell wall protein ECM33	*Saccharomyces cerevisiae*	43.8	1
89	A0A438I656	*1, 2, 4, 5, 6, 8, 9* and *10*/**A, B, C**	Glucan endo-(1,3)-*β*-glucosidase	*Vitis vinifera*	36.8	-
90	Q9S944	*1, 3* and *8*/**D**	Vacuolar invertase 1	*Vitis vinifera*	71.5	1,2,3,4,6
91	Q7XAU6	*1, 4, 5, 6, 8, 9, 10, 11, 12* and *13*/**A, B, C, D**	Class IV chitinase	*Vitis vinifera*	27.5	2,3,4,6
92	A6ZVQ6	*2*/**A, B**	Cell wall mannoprotein	*Saccharomyces cerevisiae*	26.6	-
93	A0A438I659	*1, 2, 4, 5, 6, 8, 9* and *10*/**A, B, C**	Glucan endo-(1,3)-*β*-glucosidase	*Vitis vinifera*	23.9	-
94	A0A438DX78	*4* and *5*/**A, B**	*β*-Fructofuranosidase, soluble isoenzyme I	*Vitis vinifera*	23.9	-
95	A0A438JJ75	*4, 5, 6, 8, 9, 10, 11, 12, 14* and *16*/**A, B, C, D**	Thaumatin-like protein	*Vitis vinifera*	23.9	1,2,3,4,5,6
96	A0A438BZP1	*6, 8, 9, 10, 11, 12, 13, 14* and *15*/**B, C, D**	Thaumatin-like protein	*Vitis vinifera*	36.8	1,2,3,4,5,6
97	Q756G2	*8, 9* and *14*/**C**	Probable E3 ubiquitin-protein ligase TOM1	*Ashbya gossypii*	372.2	-
98	A0A438JJ53	*8, 9, 12, 13* and *14*/**C, D**	Thaumatin-like protein	*Vitis vinifera*	23.9	1,2,3,4,5,6
99	F8KAD7	*9*/ **B**	Endo-(1,3)-*β*-glucanase	*Vitis vinifera*	34	1,2,6
100	F8KAD8	*10* and *11*/**C**	Endo-(1,3)-*β*-glucanase	*Vitis vinifera*	63.5	1,2,6
101	A0A438GZ57	*16*/ **D**	Putative non-specific lipid-transfer protein AKCS9	*Vitis vinifera*	9.8	3,4,6
102	Q850K5	*16*/**C, D**	Non-specific lipid-transfer protein	*Vitis vinifera*	11.7	3,4,6

* Ref. means References in which a protein or a similar one was identified. 1: Kwon [30]; 2: Cilindre et al. [22]; 3: Marangon et al. [38]; 4: Wigand et al. [15]; 5: D’Amato et al. [39]; 6: D’Amato et al. [12].

**Table 2 biomolecules-13-00650-t002:** Uncharacterized proteins identified by the MS-based proteomics of a Silvaner wine.

	IN-GEL (Exclusively Identified by *in-gel* Digestion)				
	Accession	Gel Band	Description	Organism	MW (kDa)
1	A0A438J4X9	1	Uncharacterized protein	*Vitis vinifera*	67.3
2	F6HUG6	1, 4 and 5	Uncharacterized protein	*Vitis vinifera*	25.3
3	A0A438HSP1	2 and 9	Uncharacterized protein	*Vitis vinifera*	32.6
4	A0A438J6G3	2	Uncharacterized protein	*Vitis vinifera*	77.5
5	A5AP16	2	Uncharacterized protein	*Vitis vinifera*	61.5
6	A0A438HTJ6	3	Uncharacterized protein	*Vitis vinifera*	26.6
7	A5B108	3	Uncharacterized protein	*Vitis vinifera*	101.2
8	A5BPD3	3	Uncharacterized protein	*Vitis vinifera*	93.1
9	A5BUH4	3 and 6	Uncharacterized protein	*Vitis vinifera*	73.7
10	D7SRI7	3	Uncharacterized protein	*Vitis vinifera*	44.4
11	A5BGP0	4	Uncharacterized protein	*Vitis vinifera*	42.1
12	A5BD73	4	Uncharacterized protein	*Vitis vinifera*	73.2
13	A5BWA5	4	Uncharacterized protein	*Vitis vinifera*	28.7
14	A5AD63	4, 9 and 13	Uncharacterized protein	*Vitis vinifera*	71.8
15	F6GZ16	5	Uncharacterized protein	*Vitis vinifera*	98.2
16	A0A438IVS9	7	Uncharacterized protein	*Vitis vinifera*	88.7
17	A5AYX1	7	Uncharacterized protein	*Vitis vinifera*	73.9
18	A5B6K0	9	Uncharacterized protein	*Vitis vinifera*	91.9
19	A5BKS0	9	Uncharacterized protein	*Vitis vinifera*	71.5
20	A5BW59	9	Uncharacterized protein	*Vitis vinifera*	91.8
21	A5BX40	9	Uncharacterized protein	*Vitis vinifera*	147.5
22	A0A1V2L6J1	9	Uncharacterized protein	*Cyberlindnera fabianii*	105.9
23	A0A438JPS2	9	Uncharacterized protein	*Vitis vinifera*	76.1
24	A5BRN8	9	Uncharacterized protein	*Vitis vinifera*	38.3
25	D7SL13	9	Uncharacterized protein	*Vitis vinifera*	6.4
26	A5AVZ0	9	Uncharacterized protein	*Vitis vinifera*	168.4
27	A5BVR4	10	Uncharacterized protein	*Vitis vinifera*	38.6
28	F6HAW3	11	Uncharacterized protein	*Vitis vinifera*	32
29	A5B6N1	11	Uncharacterized protein	*Vitis vinifera*	54.9
30	D7SVF8	12	Uncharacterized protein	*Vitis vinifera*	16.8
31	A0A438I1U6	13	Uncharacterized protein	*Vitis vinifera*	10.8
32	F6I094	13	Uncharacterized protein	*Vitis vinifera*	58.7
33	A5AK33	14	Uncharacterized protein	*Vitis vinifera*	36.1
34	A5B9R1	14	Uncharacterized protein	*Vitis vinifera*	248.6
35	A5B1A9	15	Uncharacterized protein	*Vitis vinifera*	69.3
36	A0A438JBK9	15	Uncharacterized protein	*Vitis vinifera*	24.9
37	A5BEX7	15	Uncharacterized protein	*Vitis vinifera*	118.5
38	A5BUI9	15	Uncharacterized protein	*Vitis vinifera*	40.2
39	A5CAU1	15	Uncharacterized protein	*Vitis vinifera*	84.7
40	A5AT89	16	Uncharacterized protein	*Vitis vinifera*	65.6
	**IN-SOLUTION (exclusively identified by *in-solution* digestion)**				
	**Accession**	**SEC Fraction**	**Description**	**Organism**	**MW (kDa)**
41	F6H9W6	A	Uncharacterized protein	*Vitis vinifera*	133.1
42	A5BP85	B	Uncharacterized protein	*Vitis vinifera*	113.1
43	A5BY31	C	Uncharacterized protein	*Vitis vinifera*	125.3
44	D7TT81	C	Uncharacterized protein	*Vitis vinifera*	47
45	F6H4B3	C	Uncharacterized protein	*Vitis vinifera*	58.1
46	A5BYL8	D	Uncharacterized protein	*Vitis vinifera*	103.5
47	A0A438FVB3	D	Uncharacterized protein	*Vitis vinifera*	22.2
	**IN-GEL/IN-SOLUTION (identified by *in-gel* and *in-solution* digestion)**				
	**Accession**	**Gel Band/SEC fraction**	**Description**	**Organism**	**MW (kDa)**
48	F6HMA2	*1*/ **A, B**	Uncharacterized protein	*Vitis vinifera*	60.7
49	F6HAU0	*4, 5, 6, 9, 10, 11* and *12*/**A, B, C**	Uncharacterized protein	*Vitis vinifera*	60
50	F6HUH1	*4, 5, 6, 8, 9, 10, 11, 12, 13* and *14* /**B, C, D**	Uncharacterized protein	*Vitis vinifera*	24
51	A5C9F1	*10, 11* and *16*/**A, B**	Uncharacterized protein	*Vitis vinifera*	23.8
52	D7TXF5	*10, 11* and *16*/**D**	Uncharacterized protein	*Vitis vinifera*	15.1

**Table 3 biomolecules-13-00650-t003:** Comparison of wine proteomics results in terms of wine type, methods of separation, MS analysis and protein digestion, and the number of identified proteins found in the literature.

Wine	Protein Extraction	Protein Separation	MS Analysis	Digestion Method	Identified Proteins (n)	% of Grape + Yeast Proteins	Reference
**Sauvignon blanc**	Cellulose acetate membrane (MWCO—5 kDa) Precipitation [(NH_4_)_2_SO_4_]	SDS-PAGE	Nano-LC-MS Ion trap MS	*In-gel*	**Total: 20** 5 (grape) 12 (yeast) 1 (fungi) 2 (bacteria)	85%	Kwon [30]
**Chardonnay**	Polysulfone membrane (MWCO—10 kDa) Precipitation (85%—C_2_H_6_O + 15% C_2_HCl_3_O_2_)	Isoelectric Focusing (IEF) SDS-PAGE	Nano-LC-MS/MS Ion trap MS	*In-gel*	**Total: 13** 10 (grape) 1 (yeast) 2 (fungi)	84.6%	Cilindre et al. [22]
**Semillon**	Precipitation [(NH_4_)_2_SO_4_]	Hydrophobic interaction chromatography (HIC) Reversed phase HPLC SDS-PAGE	Nano-LC-MS/MS TOF-MS	*In-gel* *In-solution*	**Total: 10** 10 (grape)	100%	Marangon et al. [38]
**German Portugieser**	Cellulose membrane (MWCO—3.5 kDa)	SDS- PAGE	LC-MS TOF-MS	*In-gel*	**Total: 18** 12 (grape) 6 (yeast)	100%	Wigand et al. [15]
**Valpolicella**	Protein adsorption (ProteoMiner beads) Protein desorption (Laemmli buffer)	SDS-PAGE	LC-MS TOF-MS	*In-gel*	**Total: 23** 1 (grape) 4 (yeast) 13 (fungi) 2 (bacteria) 3 (bovine)	17.3%	D’Amato et al. [39]
**Recioto**	Protein adsorption (ProteoMiner beads)	SDS-PAGE	Nano-LC-MS/MS	*In-gel*	**Total: 106** 95 (grape) 11 (yeast)	100%	D’Amato et al. [12]
**Silvaner**	**Ultrafiltration** **Cellulose** **membrane** **(MWCO—10 kDa)**	**Size exclusion chromatography (SEC)** **SDS-PAGE**	**LC-MS** **Quadrupole Orbitrap**	*In-gel* *In-solution*	**Total: 154** **91 (grape)** **47 (yeast)** **12 (fungi)** **4 (bacteria)**	**89.6%**	**Present study**

**Table 4 biomolecules-13-00650-t004:** Characterized proteins from a Silvaner wine identified by MS-based proteomics. The proteins are classified by cell function, organism source, and molecular mass (MW).

n°	Protein Description	Organism	MW (kDa)	Digestion Method
**Gene expression and nucleotide metabolism**				
**1**	DNA binding protein	*Geotrichum candidum*	153.2	*In-gel*
**2**	6-carboxy-5,6,7,8-tetrahydropterin synthase	*Methylobacterium* sp.	13.5	*In-gel*
**3**	Nuclear GTP-binding protein NUG1	*Pichia kudriavzevii*	58.7	*In-gel*
**4**	ATP-dependent RNA helicase DBP1	*Saccharomyces cerevisiae*	67.9	*In-gel*
**5**	Tyrosine-DNA phosphodiesterase	*Saccharomyces cerevisiae*	62.2	*In-gel*
**6**	Daughter-specific expression-related protein	*Saccharomyces cerevisiae*	121.1	*In-gel+In-solution*
**7**	Putative ribonuclease H protein	*Vitis vinifera*	16.6	*In-gel*
**8**	Retrovirus-related Pol polyprotein from transposon RE1	*Vitis vinifera*	73.7	*In-gel*
**9**	RNA exonuclease 4	*Vitis vinifera*	44.2	*In-gel*
**10**	Transposon Ty3-I Gag-Pol polyprotein	*Vitis vinifera*	59.1	*In-solution*
**11**	5′-nucleotidase SurE	*Vitis vinifera*	39.7	*In-gel*
**Metabolic breakdown and formation of carbohydrates**				
**12**	Pectin lyase A	*Aspergillus niger*	39.7	*In-gel*
**13**	Glycosidase	*S. cerevisiae x S. kudriavzevii*	53.7	*In-solution*
**14**	Endo-(1,3)-*β*-glucanase	*Saccharomyces uvarum*	34	*In-gel+In-solution*
**15**	Exo-(1,3)-*β*-glucanase of the cell wall	*Saccharomyces uvarum*	51.2	*In-gel+In-solution*
**16**	Glucan endo-(1,3)-*β*-glucosidase	*Vitis vinifera*	36.8	*In-gel+In-solution*
**17**	UDP-glycosyltransferase 85A8	*Vitis vinifera*	20.5	*In-gel*
**18**	Vacuolar invertase 1	*Vitis vinifera*	71.5	*In-gel+In-solution*
**19**	*β*-fructofuranosidase, soluble isoenzyme I	*Vitis vinifera*	63.5	*In-gel+In-solution*
**Proteins involved in post-translational modifications**				
**20**	Aminopeptidase	*Novosphingobium* sp.	72	*In-gel*
**21**	Non-specific serine/threonine protein kinase	*S. cerevisiae x S. kudriavzevii*	120	*In-gel*
**22**	Cysteine proteinase inhibitor	*Vitis vinifera*	11.2	*In-solution*
**23**	IAA-amino acid hydrolase ILR1-like 4	*Vitis vinifera*	72.7	*In-gel*
**24**	*α*-Crystallin domain-containing protein 22.3	*Vitis vinifera*	18.1	*In-gel*
**Lipid metabolism**				
**25**	Putative lipase ATG15	*Pichia kudriavzevii*	56.8	*In-gel*
**26**	Lysophospholipase	*Saccharomyces cerevisiae*	71.6	*In-solution*
**27**	Putative non-specific lipid-transfer protein AKCS9	*Vitis vinifera*	9.8	*In-gel+In-solution*
**28**	Non-specific lipid-transfer protein	*Vitis vinifera*	11.7	*In-gel+In-solution*
**Cell defense**				
**29**	Killer toxin resistant protein	*Saccharomyces cerevisiae*	30	*In-gel*
**30**	Pathogen-related protein	*Saccharomyces cerevisiae*	30.6	*In-gel*
**31**	Class IV endochitinase (Fragment)	*Vitis vinifera*	27	*In-gel*
**32**	Endochitinase EP3	*Vitis vinifera*	27.2	*In-gel*
**33**	LysM domain-containing GPI-anchored protein 1	*Vitis vinifera*	43.7	*In-gel*
**34**	Thaumatin-like protein	*Vitis vinifera*	23.9	*In-gel+In-solution*
**Cell metabolism and signaling**				
**35**	Probable E3 ubiquitin-protein ligase TOM1	*Ashbya gossypii*	372.2	*In-gel+In-solution*
**36**	Cytokinesis protein sepH	*Cyberlindnera fabianii*	116.3	*In-gel*
**37**	Protein that regulates signaling from a G protein *β* subunit Ste4p	*Kazachstania saulgeensis*	25.7	*In-gel*
**38**	ACC synthase	*Penicillium citrinum*	48.2	*In-solution*
**39**	PAS domain-containing protein	*Methylobacterium* sp.	21.3	*In-solution*
**40**	Acid phosphatase	*Saccharomyces cerevisiae*	52.9	*In-solution*
**39**	GTPase-activating protein	*Saccharomyces cerevisiae*	53.9	*In-solution*
**40**	Histidine kinase osmosensor that regulates an osmosensing MAP kinase cascade	*Saccharomyces cerevisiae*	134.5	*In-solution*
**41**	Cytochrome P450 81E8	*Vitis vinifera*	16.9	*In-gel*
**42**	Ethylene-overproduction protein 1	*Vitis vinifera*	113.4	*In-gel*
**43**	Pentatricopeptide repeat-containing protein	*Vitis vinifera*	104.7	*In-solution*
**44**	Plasma membrane ATPase	*Vitis vinifera*	105.8	*In-solution*
**45**	Rust resistance kinase Lr10	*Vitis vinifera*	68.4	*In-gel*
**46**	Ubiquitin-60S ribosomal protein L40	*Vitis vinifera*	80.1	*In-solution*
**Cell structural elements**				
**47**	Sensitive to high expression protein 9, mitochondrial	*Cyberlindnera fabianii*	42.6	*In-gel*
**48**	Component of the Sec23p-Sfb3p heterodimer of the COPII vesicle coat	*Kazachstania saulgeensis*	106.6	*In-gel*
**49**	Cell wall mannoprotein	*Saccharomyces cerevisiae*	29.6	*In-gel+In-solution*
**50**	Cell wall protein ECM33	*Saccharomyces cerevisiae*	43.8	*In-gel+In-solution*
**51**	Seripauperin	*Saccharomyces cerevisiae*	17.7	*In-solution*

## Data Availability

The original contributions presented in the study are included in the article and Appendix A. Further inquiries can be directed to the corresponding author.

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
