# Peer review of "Mass Spectrometry-Based Proteomic Profiling of a Silvaner White Wine"

_biomolecules, 2023, doi:10.3390/biom13040650_

Round 1
Reviewer 1 Report
The manuscript entitled “Mass spectrometry-based proteomic profiling of a Silvaner white wine.” Was carefully reviewed. The authors have performed mass spectrometry based proteomic profiling using in-gel and in-solution digestion approach (Bottom-up proteomics). The study is interesting in terms of proteomic profiling. Although, I suggest the authors to address the following comments to proceed further in my decision.
1. Introduction part is lengthy. Suggestable to improve the introduction part and in short manner.
2. What is the % of sequence coverage to confirm the protein from the mass spectrometry data.
3. What is the % of yield from the sample to protein fraction after FPLC fractionation.
4. Why there is much variation from in-gel to in-solution proteomics results.
5. From Fig. 1b, the proteins in-gel proteins were characterized from MS data. Have the authors tried for top-down approach to confirm the kDa of in—gel proteins using desalting technique. Since the instrument resolution is high and possibility to confirm the kDa in straight forward approach.
Author Response
We highly appreciate the opportunity to revise our manuscript submitted to the Special Issue 2nd Edition: Biochemistry of Wine and Beer. We sincerely thank the editor and all reviewers for their efforts, their constructive remarks and the positive feedback.
In the following, we have addressed the reviewers’ comments (bold) point by point. Modifications of the manuscript are given with line numbers in this letter and the given line numbers refer to the revised version of the 'tracked changes' manuscript.
Beyond the reviewers’ comments, we carried out a few minor corrections as well as typographical and editorial changes that had no influence on our results or discussion. We also updated the references. Our revisions are marked with 'tracked changes'.
Reviewer 1
Comments and Suggestions for Authors
- Introduction part is lengthy. Suggestable to improve the introduction part and in short manner.
The introduction was shortened. Two complete sentences were removed (lines 40-42 and lines 117-118) and eight sentences were modified and shortened (lines 42-46, lines 67-68, line 50-58, lines 72-73, line 84, lines 101-105, line 117-119 and lines 118-120). Moreover, some unnecessary words were removed (line 36, line 37, line 46, line 76, line 99, line 106 and lines 109-110).
- What is the % of sequence coverage to confirm the protein from the mass spectrometry data.
The % of coverage and the identified unique peptides for all the proteins are presented in the Supplementary data 1. We considered the unique peptides as reference for protein identification, as described in the section 2.5.4.
- What is the % of yield from the sample to protein fraction after FPLC fractionation.
The initial protein concentration was determined to be 0.5±0.1 mg/mL. The separated protein fractions corresponded to relative percentages of 0.8% (peak 1), 3.5% (peak 2), 92.5% (peak 3) and 3.2% (peak 4). These data were added to the supplementary information as Figure S1.
- Why there is much variation from in-gel to in-solution proteomics results.
The protein identification by using two different techniques (in-gel and in-solution) was performed in the present study to overcome the limitations of each individual technique as described in the Discussion section.
In lines 349ff “The in-gel digestion allowed the identification of proteins after an additional step of separation (gel electrophoresis) and had the advantage of reducing the mixture of proteins” and in lines 355ff “In contrast, the in-solution approach allowed the direct LC-MS/MS analysis of the digested peptide mixtures, avoiding the risk of protein losses during further fractionation steps.”
- From Fig. 1b, the proteins in-gel proteins were characterized from MS data. Have the authors tried for top-down approach to confirm the kDa of in—gel proteins using desalting technique. Since the instrument resolution is high and possibility to confirm the kDa in straight forward approach.
The in-gel analysis was performed based on bottom-up LC-MS analysis with pre-steps of protein fractionation and denaturation. We believe that top-down and native analysis of proteins in a wine is interesting to perform and can definitely be a topic for future research.
Furthermore, we have identified protein fragments from a Silvaner wine using top-down proteomics and made here a comparative analysis (data presented in Supplementary Table S1).
The study is available at https://doi.org/10.1016/j.foodchem.2021.130437
Reviewer 2 Report
Review on “Mass spectrometry-based proteomic profiling of a Silvaner white wine”
The study is well performed and well presented by the authors. As such, it deserves publication. Below are some minor suggestions to the authors.
1. Looking at the field of profiling at perspective, the introduction should make a mention to other analytical methods used for profiling, authentication. I suggest the following recent references: 1. Oliva, E.; Mir-Cerdà, A.; Sergi, M.; Sentellas, S.; Saurina, J. Characterization of Sparkling Wine Based on Polyphenolic Profiling by Liquid Chromatography Coupled to Mass Spectrometry. Fermentation 2023, 9, 223 and 2. Burns, R. L., Alexander, R., et al. (2021). A fast, straightforward and inexpensive method for the authentication of baijiu Spirit samples by fluorescence spectroscopy. Beverages, 7(3), 65.
2. In Figure 1 the writing is of low quality (resolution) compared to the text writing. If it is possible please improve.
3. Table 1 can be transferred to the supplementary material, but I will leave this to the discretion of the authors.
Again, congratulations for the fine research.
Author Response
We highly appreciate the opportunity to revise our manuscript submitted to the Special Issue 2nd Edition: Biochemistry of Wine and Beer. We sincerely thank the editor and all reviewers for their efforts, their constructive remarks and the positive feedback.
In the following, we have addressed the reviewers’ comments (bold) point by point. Modifications of the manuscript are given with line numbers in this letter and the given line numbers refer to the revised version of the 'tracked changes' manuscript.
Beyond the reviewers’ comments, we carried out a few minor corrections as well as typographical and editorial changes that had no influence on our results or discussion. We also updated the references. Our revisions are marked with 'tracked changes'.
- Looking at the field of profiling at perspective, the introduction should make a mention to other analytical methods used for profiling, authentication. I suggest the following recent references: 1. Oliva, E.; Mir-Cerdà, A.; Sergi, M.; Sentellas, S.; Saurina, J. Characterization of Sparkling Wine Based on Polyphenolic Profiling by Liquid Chromatography Coupled to Mass Spectrometry. Fermentation 2023, 9, 223 and 2. Burns, R. L., Alexander, R., et al. (2021). A fast, straightforward and inexpensive method for the authentication of baijiu Spirit samples by fluorescence spectroscopy. Beverages, 7(3), 65.
Both references were added to the discussion section (lines 323-325).
- In Figure 1 the writing is of low quality (resolution) compared to the text writing. If it is possible please improve.
The figure was modified (page 7). The quality of the text was improved.
- Table 1 can be transferred to the supplementary material, but I will leave this to the discretion of the authors.
Table 1 was kept in the main manuscript, as it summarizes our main results. Complete protein identification data is presented in Supplementary Data 1.
Round 2
Reviewer 1 Report
Authors have addressed the comments raised by the reviewer. I would suggest to accept the manuscript
Author Response
We sincerely thank the reviewer for their efforts.